# The Anti-Obesity Potential of Superparamagnetic Iron Oxide Nanoparticles against High-Fat Diet-Induced Obesity in Rats: Possible Involvement of Mitochondrial Biogenesis in the Adipose Tissues

**DOI:** 10.3390/pharmaceutics14102134

**Published:** 2022-10-08

**Authors:** Aisha H. A. Alsenousy, Rasha A. El-Tahan, Nesma A. Ghazal, Rafael Piñol, Angel Millán, Lamiaa M. A. Ali, Maher A. Kamel

**Affiliations:** 1Department of Biochemistry, Medical Research Institute, Alexandria University, 165 El-Horeya Rd, Alexandria 21561, Egypt; 2Instituto de Nanociencia y Materiales de Aragón (INMA), CSIC-Universidad de Zaragoza, 50009 Zaragoza, Spain; 3IBMM, University Montpellier, CNRS, ENSCM, 34093 Montpellier, France

**Keywords:** obesity, superparamagnetic iron oxide nanoparticles, white adipose tissue browning, mitochondrial biogenesis, mitochondrial DNA copy number

## Abstract

Background: Obesity is a pandemic disease that is rapidly growing into a serious health problem and has economic impact on healthcare systems. This bleak image has elicited creative responses, and nanotechnology is a promising approach in obesity treatment. This study aimed to investigate the anti-obesity effect of superparamagnetic iron oxide nanoparticles (SPIONs) on a high-fat-diet rat model of obesity and compared their effect to a traditional anti-obesity drug (orlistat). Methods: The obese rats were treated daily with orlistat and/or SPIONs once per week for 8 weeks. At the end of the experiment, blood samples were collected for biochemical assays. Then, the animals were sacrificed to obtain white adipose tissues (WAT) and brown adipose tissues (BAT) for assessment of the expression of thermogenic genes and mitochondrial DNA copy number (mtDNA-CN). Results: For the first time, we reported promising ameliorating effects of SPIONs treatments against weight gain, hyperglycemia, adiponectin, leptin, and dyslipidemia in obese rats. At the molecular level, surprisingly, SPIONs treatments markedly corrected the disturbed expression and protein content of inflammatory markers and parameters controlling mitochondrial biogenesis and functions in BAT and WAT. Conclusions: SPIONs have a powerful anti-obesity effect by acting as an inducer of WAT browning and activator of BAT functions.

## 1. Introduction

Obesity is an increasingly spreading pandemic that is a global health issue and has a direct economic effect on healthcare services. Its prevalence has tripled globally since 1975, according to the WHO. More than 1.9 billion (39%) adults were estimated to be overweight, with 650 million obese, accounting for nearly 13% of the world’s adult population [1]. Obesity is caused by a variety of factors, including, but not limited to, genetic, epigenetic, biochemical, hormonal, microbial, sociocultural, and environmental influences that disrupt the balance between calorie intake and energy expenditure [2]. Frequently, it is associated with several disorders such as type 2 diabetes (T2D), insulin resistance (IR), cardiovascular disorders, and cancers [3]. It is characterized by a state of chronic low-level inflammation due to the increased expression level of tumor necrosis factor alpha (TNF-α) from adipose tissue, which participates in the simulation of lipolysis in adipocytes and in insulin resistance development [4,5].

Adipose tissue is one of the most essential organs affected by obesity. It is divided into white adipose tissue (WAT) and brown adipose tissue (BAT). The WAT is composed of large lipid droplets and participates in energy storage as triglycerides (TG), whereas BAT has high mitochondrial content and involves energy consumption via non-shivering thermogenesis, mostly through tissue-specific uncoupling protein-1 (UCP-1) [6]. Mitochondria are key organelles that control the physiological roles of adipocytes such as regulation of whole-body energy homeostasis, adipocyte differentiation, lipid homeostasis, insulin sensitivity, oxidative capacity, and browning of WAT into beige adipose tissue through the transcriptional control of the brown fat gene program (e.g., UCP-1) [7]. Beige adipocytes are characterized by possessing more mitochondria than WAT, with enhanced gene expression of proteins involved in lipolysis and thermogenesis. So, WAT browning and/or BAT activation constitute a possible clinical target for the treatment of obesity [8].

Mitochondrial biogenesis is controlled by the transcription factor peroxisome proliferator-activated receptor-gamma coactivator-1alpha (PGC-1α) [9]. Sirtuin-1 (SIRT-1) is an important regulator of adipocyte differentiation and adipogenesis, and it is downregulated by a high-fat diet (HFD) in adipose tissue [10,11]. SIRT-1 induces mitochondrial biogenesis by activating PGC-1α [12]. Sterol regulatory element-binding protein 1c (SREBP-1c), a master regulator of fatty acid (FA) biosynthesis, is upregulated in WAT metabolic dysfunction in obesity [13]. The mitochondrial DNA copy number (mtDNA-CN) reflects the level of mtDNA in a cell relative to the nuclear DNA (nDNA) and is linked to mitochondrial enzyme activity and ATP level, all of which are considered indicators of mitochondrial biogenesis and function [14].

Currently, anti-obesity or weight loss treatments decrease or regulate weight by modifying calorie absorption or appetite; for example, orlistat serves as an antagonist of the lipase enzyme, which inhibits TG from being digested, thus inhibiting TG absorption and hydrolysis. These therapies are only prescribed for short-term consumption, making them ineffective for chronically obese patients who will need to lose weight over months. As a result, researchers are looking for new approaches to increase thermogenesis and treat obesity and its associated health risks [15]. Nanotechnology is a branch of science that involves design and synthesis of nano-sized materials (1 to 100 nm) for application in various fields such as medicine, and it is regarded as a promising approach in obesity treatment [16].

Superparamagnetic iron oxide nanoparticles (SPIONs) are inorganic nanomaterials that show special properties such as superparamagnetism and low toxicity. SPIONs are used in a variety of biomedical applications either as a therapeutic, diagnostic, or theranostic tool for hyperthermia, drug delivery, magnetic resonance imaging, and cell separation [17]. Over several years, we have been developing SPIONs for biomedical applications, and they showed a non-specific anticoagulant effect with no hemolytic effect on blood, low cytotoxicity, and powerful diagnostic ability in magnetic resonance imaging in vivo [18,19].

Recently, our lab indicated that SPIONs have an anti-diabetic effect on the diabetic rat model, with a low toxic effect recorded for the dose (22 μmol Fe/kg) [20]. Sharifi et al. showed the involvement of SPIONs in the regulation of genes involved in lipid and glucose metabolism, suggesting that they could be used as therapeutics for diabetes and obesity [21].

Therefore, our study aims to explore the anti-obesogenic potential of SPIONs compared to the commercial orlistat in the rat model of obesity. In this study, the effect of SPIONs coating will be evaluated by using two different types of SPIONs coated with different molecular weights of the polyethylene glycol (PEG) (550 and 2000 Da). The SPIONs will be used alone or combined with orlistat. Finally, we will explore the possible molecular mechanisms of SPIONs’ effects including inflammation, lipogenesis, IR, mitochondrial biogenesis, and WAT browning.

## 2. Materials and Methods

### 2.1. Synthesis of Ferrofluids and Characterization

The synthesis was performed in two steps: (a) coating preparation and (b) synthesis of SPIONs coated with PEG (Mw: 550) (SPION-PEG-550) or with PEG (Mw: 2000) (SPION-PEG-2000), as previously described in [20]. The prepared ferrofluids were purified by filtration through 0.45 and 0.22 µm nitrocellulose membrane filters (Merck Millipore Ltd., Carrigtwohill, County Cork, Ireland)) followed by magnetic separation using MidiMACS^TM^ separator and LS column (Miltenyi Biotec GmbH, Bergisch Gladbach, Germany). The collected nanoparticles were dispersed in MilliQ water, and the pH was adjusted to physiological pH. Samples were sterilized by filtration through Millex^®^-GP sterile syringe filters (Merck Millipore Ltd., Carrigtwohill, County Cork, Ireland) with a 0.22 µm pore size and hydrophilic polyethersulfone (PES) membrane in a laminar flow hood and stored at room temperature till the moment of animal injection. It is worth mentioning that the nanoparticle samples used in this work are the same used in [20]. A detailed and full description of the samples was included in this reference.

In order to determine the iron content, inductively coupled plasma optical emission spectrometry (ICP-OES) in a plasma 40 ICP Perkin-Elmer spectrometer was used. Hydrodynamic diameter was determined by dynamic light scattering (DLS) measurements performed using a Malvern Zetasizer NS (Malvern Instruments Ltd., Worcestershire, UK) equipped with a HeNe laser (633 nm). Zeta-potential measurements were performed using a folded capillary cell, DTS 1060 (Malvern Instruments Ltd. Worcestershire, UK). The aqueous colloidal suspension stability was verified using DLS. Transmission electron microscopy (TEM) and high-resolution (HR) TEM images were obtained in an aberration-corrected transmission electron microscope Tecnai Titan.

Structural characterization by attenuated total reflectance Fourier transform infrared spectroscopy (ATR-FTIR) and thermal analysis (TA) to confirm the attachment of bis(phophonic) end-capped PEG chains to the iron oxide nanoparticles was performed on lyophilized samples. ATR-FTIR spectra of SPIONs and SPION-PEG-NPs were recorded with a Perkin Elmer Spectrum 100 FTIR spectrophotometer equipped with a UATR sampling accessory in the range 4000–380 cm^−1^. Thermogravimetric (TGA) analysis was conducted on a TA Instruments SDT 2960 simultaneous DTA-TGA. Samples were heated from 25 to 700 °C at a heating rate of 10 °C·min^−1^ under air flow. The mass remaining at 700 °C was taken as the fraction of maghemite present in the nanoparticles.

### 2.2. Experimental Animals

A total number of 56 albino Sprague-Dawley male rats, 2 months old (80–90 g), was used. The animals were obtained from the animal house of Medical Research Institute, Alexandria University, Egypt. Rats were housed in standard cages in a well-ventilated room (25 ± 2 °C), with a relative humidity of (43 ± 3), with free access to water and food and 12 h light/dark cycle before experimentation.

### 2.3. Ethical Statement

All experiments pursued the standards of the National Institutes of Health’s *Guide for the Care and Use of Laboratory Animals* (NIH, Bethesda, MD, USA, publications no. 8023, revised 1978) and were performed after the approval of the Institutional Animal Care and Use Committee (IACUC), Alexandria University, Egypt (approval no. AU01219101613). The study also followed ARRIVE guidelines and complied with the National Research Council’s guide for the care and use of laboratory animals.

### 2.4. Obesity Induction

Obesity was induced in rats by feeding them with an obesogenic diet for 2–3 months. Rats that became 20% heavier than control rats of the same age were considered obese. The composition of the obesogenic diet used in this experiment (per 100 g diet) was 30 g protein (300 cal), 26.5 g fat (195 cal lard, 70 cal corn oil), 36.5 g carbohydrate (105 cal dextran, 106 cal corn starch, 140 cal sucrose), 3 g vitamin mix (30 cal), and 4 g mineral mix (40 cal) [22].

### 2.5. Experimental Design

Animals were classified into the following groups: (1) healthy control group that consisted of 8 healthy male rats, after the establishment of obesity, with the 48 obese male rats being divided into six groups (8 rats each) according to the treatment; (2) untreated obese group, (3) orlistat-treated obese group that was orally treated with orlistat (Orly^R^ from EVA PHARMA Product Code: 11659) dissolved in dimethyl sulfoxide at a dose of 30 mg/kg daily [23]; (4) SPION-PEG-550-treated obese group, in which obese rats were intravenously injected with SPION-PEG-550 at a dose of 22 μmol Fe/kg once a week [20,23]; (5) SPION-PEG-550 + orlistat-treated obese group, in which obese rats were intravenously injected with SPION-PEG-550 at a dose of 22 μmol Fe/kg once a week and were orally treated with orlistat at a dose of 30 mg/kg daily; (6) SPION-PEG-2000-treated obese group, in which obese rats were intravenously injected with SPION-PEG-2000 at a dose of 22 μmol Fe/kg once a week [20,23]; (7) SPION-PEG-2000 + orlistat-treated obese group, in which obese rats were intravenously injected with SPION-PEG-2000 at a dose of 22 μmol Fe/kg once a week and were orally treated with orlistat at a dose of 30 mg/kg daily.

All treatments were continued for 8 weeks, and all obese rats were maintained under the obesogenic diet during the experimental period.

### 2.6. Collection of Samples

After the end of the treatment period, overnight fasting rats were weighed and fasting blood glucose (FBG) was determined in the fasted animals with an automatic glucose meter (Accu-Chek, Roche Diagnostics, Mannheim, Germany) using blood samples from the tail tip. Afterwards, rats were anesthetized by intraperitoneal injection of ketamine (75 mg/kg) and xylazine (10 mg/kg) and then sacrificed. The serum samples were prepared by collecting the blood from the retroorbital vein in anticoagulant free tubes, followed by centrifugation at 3000× *g* for 10 min. The serum samples were used for the determination of insulin, lipid profile (TG, total cholesterol (TC), high-density lipoprotein cholesterol (HDL-C), low-density lipoprotein cholesterol (LDL-C)), alanine aminotransferase (ALT) activity, aspartate aminotransferase (AST) activity, urea, creatinine, leptin, adiponectin, and non-esterified fatty acid (NEFA) levels. The WAT and BAT were obtained and divided into three aliquots: (i) for the extraction of total RNA for quantitative real-time-polymerase chain reaction (qRT-PCR) analysis, in order to assess the gene expression of TNF-α, PGC-1α, SIRT-1, SREBP-1c, and UCP-1, (ii) for the extraction of total DNA for the determination of mtDNA-CN, and (iii) for protein assays.

### 2.7. Serum Parameters Measurements

Serum insulin concentration was determined following the instructions of the Insulin rat ELISA kit (EMD Millipore, Burlington, MA, USA), absorbance was measured at 450 nm, and the homeostasis model assessment index for insulin resistance (HOMA-IR) was then calculated using the following formula [24]:$$\mathrm{HOMA}-\mathrm{IR}=\frac{\mathrm{Fasting}\;\mathrm{insulin}\left(\left(µ\mathrm{IU}\right)/\mathrm{mL}\right)\times \;\mathrm{Fasting}\;\mathrm{glucose}\left(\mathrm{mg}/\mathrm{dL}\right)}{22.5\;\times \;18}$$

Serum TG, TC, and HDL–C levels were determined by the enzymatic colorimetric method using reagents obtained from BioMed Diagnostics, Inc. (White City, OR, USA), and absorbance was measured at 546 nm. Serum LDL-C was calculated from TG, TC, and HDL-C concentrations using the following equation [25]:**LDL-C (mg/dL) = TC − (HDL-C) − TG/5**

Serum ALT and AST activities were determined using reagents obtained from BioMed Diagnostics, Inc. (USA), and absorbance was measured at 340 nm. Urea and creatinine were determined using reagents obtained from BioMed Diagnostics, Inc. (USA), and absorbance was measured at 570 nm and 510 nm, respectively. Serum leptin was assayed using rat ELISA kit (eBioscience, San Diego, CA, USA), adiponectin and NEFA were assayed using rat ELISA kit (Elabscience, Houston, TX, USA), and serum lipase activity was assayed using colorimetric kit (Spectrum, Alexandria, Egypt). All procedures were performed according to the manufacturer’s instructions.

### 2.8. Mitochondrial DNA Copy Number Determination

The qRT-PCR assay was used for the determination of mtDNA number relative to nDNA. First, the total DNA was extracted from WAT and BAT using DNeasy Mini Kit (Qiagen, Hilden, Germany) according to the manufacturer’s instructions, and then the PCR reaction was performed using a specific primer pair for mtDNA sequence and a primer pair specific for nuclear sequence (PGC-1α) to perform the same number of PCR cycles and calculate the threshold cycle (Ct) of both mtDNA and nDNA sequences. The nuclear gene was used to quantify nDNA and therefore normalization of the mtDNA amount per the nDNA of the diploid cells using the equation:**R = 2^−ΔCt^ where ΔCt = Ct_mtDNA_ − Ct_nuclear_**

A specific primer pair for mtDNA (forward: 5’-ACACCAAAAGGACGAACCTG-3’; reverse: 5’-ATGGGGAAGAAGCCCTAGAA-3’) and a primer pair for the nuclear PGC-1α gene (forward: 5’-ATGAATGCAGCGGTCTTAGC-3’; reverse: 5’-AACAATGGCAGGGTTTGTTC-3’) were used. PCR reactions were carried out using Rotor Gene SYBR Green PCR Kit (Qiagen^®^, Germantown, MA, USA), 0.5 µM forward and reverse primer, and 50 ng of extracted DNA under the following conditions: 95 °C for 10 min followed by 40 cycles of 95 °C for 15 s, 60 °C for 30 s, and 72 °C for 30 s [26].

### 2.9. Gene Expression Detection of TNF-α, PGC-1α, UCP-1, SIRT-1, and SREBP-1c

Total RNA was isolated from WAT and BAT using RNeasy Mini Kit (Qiagen^®^, Germany) according to the manufacturer’s instructions, and the concentration and integrity of extracted RNA were checked using nanodrop. Reverse transcription was conducted using miScript II RT Kit according to the manufacturer’s instructions. The tissue expression of TNF-α, PGC-1α, SIRT-1, SREBP-1c, and UCP-1 was quantified in the cDNA using Rotor Gene SYBR Green PCR Kit (Qiagen^®^, USA). Quantitative PCR amplification conditions were adjusted as an initial denaturation at 95 °C for 10 min and then 45 cycles of PCR for amplification as follows: denaturation at 95 °C for 20 s, annealing at 55 °C for 20 s, and extension at 70 °C for 15 s. The housekeeping gene glyceraldehyde 3-phosphate dehydrogenase (GAPDH) was used as a reference gene for normalization. The primers used for the determination of rat genes are presented in Table 1. The relative change in mRNA expression in samples was estimated using the 2^−ΔΔCt^ method [27].

### 2.10. Protein Levels Determination of PGC-1α, SREBP-1c, and TNF-α by ELISA

The excised WAT and BAT were homogenized in bicinchoninic acid (BCA) using BCA protein assay kit (Chongqing Biospes Co., Ltd., Chongqing, China, catalog no. BWR1023) according to the instructions of the manufacturer. The supernatants were used for determination of PGC-1α using specific rat ELISA kits (MyBioSource, San Diego, CA, USA, catalog no. MBS27063799) according to the instructions of the manufacturer. Moreover, SREBP-1c and TNF-α were assayed using specific rat ELISA kits (Chongqing Biospes Co., Ltd., catalog no. BYEK3082 and BEK1214) according to the manufacturer’s instructions.

### 2.11. Statistical Analysis

Data were analyzed using SPSS software package version 18.0 (SPSS Chicago, IL, USA). The data were expressed as mean ± standard deviation (SD) and analyzed using one-way analysis of variance (ANOVA) to compare between different groups. The *p*-value was assumed to be significant at *p* < 0.05. The correlation coefficients (r) between different assayed parameters were evaluated using the Pearson correlation coefficient; *p* < 0.05 was considered as the significance limit for all comparisons.

## 3. Results

### 3.1. Ferrofluids Characterization

A detailed and full description of samples characterization is included in reference [20]. The DLS measurements showed that the hydrodynamic size values of SPION-PEG-550 and SPION-PEG-2000 were 30.1 ± 9.1 nm and 34.2 ± 10.4 nm with polydispersity index (PDI) values of 0.158 and 0.143, respectively. The aqueous colloidal suspension of ferrofluids showed great stability over time, up to several years, without any appreciable change in the stability, as was confirmed by DLS (Figure 1). After 2 years, the hydrodynamic size values of SPION-PEG-550 and SPION-PEG-2000 were 30.2 ± 8.9 nm and 35.9 ± 10.6 nm with PDI values of 0.154 and 0.130, respectively.

The TEM images (Figure 2) showed the polynuclear character of the maghemite (γ-Fe_2_O_3_) nuclei, formed by clusters with a discrete number of maghemite nanoparticles, with a spherical shape and a mean diameter of DTEM(SD) = 11.2 (2.4) nm.

The FTIR spectra confirmed the presence of the maghemite SPIONs and the PEG polymer layer around the magnetic core present in both samples (SPION-PEG-550 and SPION-PEG-2000), as shown Figure 3. The infrared spectrum of sample SPION-PEG-2000 showed a higher intensity of the characteristic band of PEG at 1105 cm^−1^ attributed to the C-O-C stretching vibration band of PEG. These data are consistent with the presence of polymer chains of higher molecular weight and a higher content in organic polymer in the SPION-PEG-2000 sample and are in concordance with the data obtained by TGA. According to the thermograms obtained, the calculated mass of PEG present in the samples was 13% for sample SPION-PEG-2000 and 6% for sample SPION-PEG-550.

### 3.2. Weight Change

Before the start of treatments, all the obese rats were significantly heavier than the control rats, with no significant difference between the obese groups. After the treatments, all the obese groups were still significantly heavier than the healthy control group; however, their body weight was significantly lower than the untreated obese rats (Table 2). The untreated obese rats and orlistat-treated rats had significantly higher weight gain compared with the healthy control rats, while the other treated obese rats had significantly lower weight gain compared with untreated obese rats. The obese rats treated with a combination of SPION-PEG-550 and orlistat showed the best lowering effect on weight gain, as shown in Table 2.

### 3.3. Parameters of Glucose Homeostasis

Untreated obese rats had a significant elevation in glucose homeostasis parameters (FBG, insulin, and HOMA-IR) compared with the healthy control group. The orlistat treatment did not significantly affect these parameters, except for HOMA-IR, which showed significant reduction compared with the untreated obese rats. The treatment of obese rats with the two types of SPIONs (SPION-PEG-550 or SPION-PEG-2000) alone or in combination with orlistat significantly reduced these parameters compared with the untreated rats with the exception of insulin which showed no significant changes with SPIONs alone. Better effects were observed in the obese rats treated with SPION-PEG-2000 combined with orlistat (Table 3).

### 3.4. Liver and Kidney Function Tests

The untreated obese rats showed significantly higher ALT and AST activities compared with healthy control rats. Orlistat-treated rats had a significant decline in both ALT and AST activities compared with the untreated rats. Moreover, the obese rats treated with both types of SPIONs showed significantly lower activities compared with obese untreated rats, especially in the rats treated with a combination of SPIONs and orlistat (Table 4).

Untreated obese rats had a mild but significant increase in urea and creatinine levels compared with healthy control rats. The group that was treated with orlistat, treated with the two different coatings of SPIONs alone, or in combination with orlistat experienced no significant changes on urea and creatinine levels compared with the untreated group (Table 4).

### 3.5. Serum of Lipid Profile

The levels of TG and total and LDL cholesterol were significantly higher while HDL cholesterol was significantly lower in the untreated obese rats compared with the healthy control group. The obese rats treated with orlistat showed significantly lower TG and total and LDL cholesterol and significantly higher HDL cholesterol levels compared with the untreated group. Moreover, the obese rats treated with the SPIONs with two different coatings showed significant improvement of lipid profile but to a lesser extent than with orlistat. The rats treated with a combination of SPIONs and orlistat showed better improvements than orlistat alone, especially those treated with SPION-PEG-2000 combined with orlistat. A similar pattern of change was observed in the levels of serum NEFA (Table 5).

### 3.6. Serum Leptin and Adiponectin Levels

The untreated obese rats showed significantly higher leptin levels than the healthy control rats. The orlistat treatment did not show significant correction of leptin level; however, the obese rats treated with SPIONs alone or in combination with orlistat showed significantly lower leptin levels compared with untreated rats and orlistat-treated rats. The best leptin-lowering effect was shown in the obese rats treated with SPION-PEG-550 combined with orlistat, but the levels of leptin were still higher than the healthy control value, as presented in Figure 4A.

The adiponectin levels showed a significant decline in all obese rats compared with healthy control rats. However, the obese rats treated with SPIONs alone or in combination with orlistat showed significantly higher adiponectin levels compared with the untreated rats. The combined treatments have the best amelioration effects on the adiponectin levels, as shown in Figure 4B.

### 3.7. TNF-α Expression in WAT and BAT

The untreated obese rats had marked upregulation of TNF-α expression at mRNA and protein levels in both WAT and BAT compared with the healthy control group. On the other hand, orlistat-treated rats showed significant downregulation of TNF-α expression at mRNA and protein levels compared with untreated obese rats in the WAT, while in BAT the expression is downregulated only at the protein level. The obese rats treated with SPIONs showed significantly downregulated expression of TNF-α at mRNA and protein in both tissues compared with untreated obese rats. The combined treatment showed more reduction in the expression of TNF-α expression at mRNA and protein levels in both tissues compared with untreated obese rats or with other treated groups (Figure 5A,B).

### 3.8. PGC-1α Expression in WAT and BAT 

The expression of PGC-1α at mRNA and protein levels of untreated obese rats showed significant downregulation in both WAT and BAT compared with healthy control rats. In WAT, only the combined treatment with SPION-PEG-2000 and orlistat showed significant upregulation and completely normalized the expression of PGC-1α. In BAT, all treatments significantly upregulated the expression of PGC-1α at mRNA and protein levels with the best effects observed in the obese rats treated with combined treatment of SPION-PEG-2000 and orlistat (Figure 6A,B).

### 3.9. SREBP-1c Expression in WAT and BAT

In both WAT and BAT, the untreated obese rats had a significant upregulation of SREBP-1c expression at mRNA and protein levels compared with the healthy control group. In WAT, all treatments significantly downregulated the expression of SREBP-1c compared with the untreated rats; however, the best effects were observed in the rats treated with a combined treatment of SPION-PEG-2000 and orlistat, which completely normalized the expression at mRNA and protein levels. Like WAT, the SREBP-1c expression in BAT was significantly downregulated by all the treatments used compared with the untreated rats, with the best effects observed in the rats treated by the combined treatments SPION-PEG-550 or SPION-PEG-2000 with orlistat (Figure 7A,B).

### 3.10. SIRT-1 Expression in WAT and BAT

The mRNA expression of SIRT-1 was significantly downregulated in both WAT and BAT of the untreated obese rats compared with healthy control rats. The orlistat treatment did not significantly affect the expression of SIRT-1 in WAT or BAT. However, the treatments with the two types of SPIONs alone significantly upregulated the expression of SIRT-1 compared with untreated rats in both tissues. In WAT, the expression of SIRT-1 was significantly upregulated by SPIONs treatment when compared with orlistat. The combined treatment of obese rats with any of SPIONs (SPION-PEG-550 or SPION-PEG-2000) together with the orlistat significantly upregulated the expression compared with the other treatments, with the best effect observed in the rats treated with SPION-PEG-2000 and orlistat, which showed complete normalization, with no significant difference from healthy controls, of SIRT-1 expression in both WAT and BAT (Figure 8).

### 3.11. UCP-1 Expression in WAT and BAT

The expression of UCP-1 was significantly downregulated in BAT of the untreated obese rats with no significant changes in WAT compared with healthy control rats. The orlistat treatment did not significantly affect the expression of UCP-1 in WAT but significantly upregulated its expression in BAT. In WAT, the treatments with the two types of SPIONs alone significantly upregulated the expression of UCP-1 compared with untreated rats, orlistat-treated obese rats, or healthy control rats. The combined treatment of SPION-PEG-550 or SPION-PEG-2000 together with the orlistat significantly upregulated the expression compared with the other treatments. In BAT, the treatment of obese rats with the SPIONs alone or in combination with orlistat showed a significant upregulation of UCP-1 expression compared with untreated obese rats. The combined treatments completely normalized the expression of UCP-1 with no significant difference observed compared with healthy controls (Figure 9).

### 3.12. Mitochondrial DNA Copy Number in WAT and BAT

In WAT, no significant difference was observed between the untreated obese rats and healthy control rats regarding the mtDNA-CN, and the treatment with orlistat did not significantly affect it. However, the treatment of obese rats with SPION-PEG-550 or SPION-PEG-2000 alone significantly increased the mtDNA-CN compared with the healthy control, untreated obese, and orlistat-treated groups. The combined treatments showed significantly higher mtDNA-CN compared with all other groups and showed about double the control value (Figure 10).

In BAT, the untreated obese rats showed a decline in the mtDNA-CN compared with the healthy control rats. Orlistat-treated rats showed significant elevation of mtDNA-CN compared to untreated obese rats. The treatment with SPIONs alone or in combination with orlistat showed a significantly higher mtDNA-CN compared with untreated obese rats and orlistat-treated rats, with the best effect observed in the combined treatments (Figure 10).

### 3.13. Correlation Studies

The statistical analysis using Pearson correlation is presented in Table 6, and the analyses showed the following:PGC-1α expression was positively correlated with UCP-1 expression in both WAT and BAT. In BAT, PGC-1α expression was positively correlated with SIRT-1 expression and mtDNA-CN. On the other hand, in WAT, PGC-1α expression was negatively correlated with SREBP-1c expression, TNF-α expression, and NEFA level.SIRT-1 expression was positively correlated with UCP-1 expression and mtDNA-CN in both tissues. However, it was negatively correlated with SREBP-1c expression and TNF-α expression in WAT and BAT, whereas in BAT, SIRT-1 expression was negatively correlated with NEFA level.Serum leptin level was positively correlated with TNF-α expression, SREBP-1c expression, and NEFA level in WAT and BAT. However, it was negatively correlated with UCP-1 expression, SIRT-1 expression, and mtDNA-CN in both organs.UCP-1 expression was positively correlated with mtDNA-CN in these tissues but was negatively correlated with TNF-α expression in WAT and BAT and negatively correlated with NEFA level.mtDNA-CN was negatively correlated with NEFA level in both WAT and BAT. On the other hand, it was negatively correlated with TNF-α expression in WAT and BAT.

## 4. Discussion

The present study showed for the first time the potential anti-obesity properties of SPIONs in an HFD rat model. This effect may be mediated through suppression of WAT expansion, induction of WAT browning, and activation of BAT function.

The HFD-obese rats developed the classical picture of obesity: they were 70% heavier than the control rats, and the weight gains during the experimental period were more than three times the control rats. Moreover, they developed hyperglycemia and insulin resistance, besides elevated liver enzyme activities (AST, ALT) and significantly higher urea and creatinine levels, though within the normal range. The transition from a metabolically stable condition to an obese and insulin-resistant state is characterized by a vicious loop that includes hyperinsulinemia, inflammation, glucose tolerance, dyslipidemia, IR, and adipose tissue expansion. Furthermore, the circulating NEFA levels in the obese rats were markedly higher than the controls, which may be due to the release of NEFA from the enlarged adipose tissue and reduced clearance [28]. The NEFA levels were positively correlated with the leptin level and negatively correlated with mtDNA-CN and with the expression of PGC-1α, SIRT-1, and UCP-1 in BAT and WAT. These patterns of correlations put the elevated NEFA in the core mechanism of obesity pathogenesis. The elevated NEFA levels induce insulin resistance and inhibit insulin’s antilipolytic action, which will increase the rate at which NEFA is released into the circulation [29]. Moreover, the elevated NEFA activated the proinflammatory pathways [30] and resulted in increased proinflammatory cytokines expression as TNF-α, IL-1b, and IL-6 [31]. All of these make NEFA the primary link between high-fat feeding and the development of inflammatory alterations [32].

In obesity, WAT expansion leads to a significant decrease of serum adiponectin levels and an increase in leptin levels that are correlated with insulin resistance [13]. Leptin inhibits appetite and food intake, stimulates energy expenditure, and also has pro-inflammatory effects contributing to the low-grade chronic inflammation by enhancing the TNF-α and IL-6 production [33] and vice versa TNF-α stimulated leptin secretion from adipocytes [34] that induces obesity [35,36]. Our study confirmed the increased levels of serum leptin and TNF-α expression in WAT and BAT, and the correlation studies indicated a positive correlation between the leptin levels and TNF-α expression in BAT, which may explain the impairment of functions of BAT in energy expenditure.

The metabolic and adipocytokine derangements in obese rats are associated with marked activation of the lipogenic protein SREBP-1c and marked suppression of the expression of genes encoding essential proteins implicated in adipose tissue differentiation and activation, as well as mitochondrial biogenesis and function (PGC-1α, UCP-1, and SIRT-1). Mitochondria play an important function in the maintenance of energy homeostasis in metabolic tissues, particularly adipose tissues. Mitochondria play an important role in adipocyte biology and growth, including adipogenesis, lipid metabolism, and thermogenesis [37,38]. Furthermore, adipocyte mitochondria can regulate whole-body energy homeostasis, insulin sensitivity, and glucose metabolism or the crosstalk between muscles and adipose tissues [39,40]. 

PGC-1α is the key transcription factor that regulates mitochondrial biogenesis and functions by controlling the expression of nuclear respiratory factor 1 (NRF-1), nuclear factor erythroid 2-related factor 2 (NRF-2), and mitochondrial transcription factor A (Tfam) [41,42]. Moreover, PGC-1α has generally been recognized as a master regulator thermogenic gene programmed in differentiated brown and beige adipocytes [43]. So, PGC-1α is essential for thermogenic adipocytes (BAT) to perform their functions, and the observed suppression of PGC-1α expression in BAT impairs their proper functions. PGC-1α is a key regulator of brown adipogenesis by helping peroxisome proliferator-activated receptor gamma (PPAR-γ) induce WAT browning. PGC-1α deficiency can cause the downregulation of UCP-1 and block mitochondria biogenesis [44]. So, the suppressed PGC-1α expression could explain the marked suppressed expression of UCP-1 in BAT found in the obese rats in the present study.

Uncoupling protein 1, a mitochondrial protein, plays a major role in the thermogenic function of BAT [45]. The activity of UCP-1 and thermogenesis in mouse BAT is correlated with body-weight control and energy homeostasis [46]. In line with our data, the UCP-1 expression is reduced in obese subjects, and the metabolic complications are improved with the pharmacological activation of UCP-1 [47]. In human adipose tissues, the expression of UCP-1 was significantly negatively correlated with fasting glucose, and TG was positively correlated with adiponectin. The visceral obesity was aggravated when UCP-1 expression was downregulated [6].

The suppressed expression of PGC-1α in obese rats was associated with a significant decline in mtDNA-CN in adipose tissues, especially BAT, which may indicate impaired mitochondrial biogenesis, while the suppression of UCP-1 in BAT impairs the mitochondrial thermogenesis and functions. The correlation studies indicated a causality relationship between the suppression of PGC-1α and downregulation of UCP-1 expression and the decline in mtDNA-CN in BAT. The impaired mitochondrial function and biogenesis in adipocytes can affect whole-body energy dysregulation and insulin resistance.

The HFD-obese rats showed significant downregulation of SIRT-1 expression and upregulation of expression and protein level of SREBP-1c compared with control rats in both BAT and WAT. SIRT-1 is known to activate the AMP-activated protein kinase (AMPK) signaling pathway and initiate the lipolysis of adipocytes and activate the thermogenic genes UCP-1 and PGC-1α [48,49]. PGC-1α then upregulates the gene expression of various key enzymes for beta-oxidation and induces fatty acid oxidation. Moreover, SIRT-1-mediated deacetylation of PPAR-γ is necessary for the transcriptional activity [50]. So, SIRT-1, AMPK, PPAR-γ, and PGC-1α cross-regulate each other in energy metabolism [51,52]. The suppressed expression of these machinery genes results in inhibited energy expenditure due to WAT expansion and impaired BAT functions. The inverse association between obesity and active BAT mass was previously confirmed [53,54]. SREBP-1c mediated de novo lipogenesis is an important nutritional regulator in the biosynthesis of FAs and triglyceride, and it also significantly correlated with both HOMA and serum insulin levels and pro-lipogenic factors [55]. 

The current approaches for obesity treatment include diet control, physical activity, drug therapy, and surgery [56]. However, the applied anti-obesity therapies have shown several limitations. Today, the modulation of mitochondrial biogenesis and activity in adipose tissues and induction of WAT browning has been proposed as a promising approach for the prevention and management of obesity by increasing the energy expenditure strategy [8]. The current study revealed for the first time the promising effects of SPIONs as an anti-obesity treatment that outperforms the commonly prescribed medication orlistat.

SPIONs treatments at the weekly i.p. dose of 22 µmol Fe/kg significantly declined the final body weights and weight gains in the obese rats during the experimental period, irrespective of the coating (PEG-550 or 2000 Da). Moreover, SPIONs treatment significantly ameliorates hyperglycemia, insulin resistance, dyslipidemia, leptin, adiponectin, and NEFA. The weekly dose of SPIONs has similar or even better effects than those observed with the daily orlistat treatment. The combined SPIONs and orlistat treatments showed more pronounced ameliorative effects, with the best outcomes observed in the obese rats treated with the weekly SPION-PEG-2000 and daily orlistat, which nearly normalized most of the studied metabolic and molecular derangement.

SPIONs treatments significantly decreased the elevated levels of leptin and NEFA in obese rats and significantly increased the level of adiponectin. The effect of SPIONs on leptin level and adiponectin level was significantly better than the effect of orlistat, which may imply a leptin-sensitizing effect of SPIONs, especially those coated with PEG-2000. Moreover, the anti-obesity action of SPIONs may be partially mediated through its lipotropic effect, as it significantly ameliorates the lipid profile like orlistat or even better. Considering SPIONs’ effect on the lipid components, it can be suggested as a potential hypolipidemic agent, which will be of great advantage for obesity. This effect of SPIONs may be a consequence of the corrected glucose homeostasis and insulin resistance; however, such effect needs further investigation

At the molecular level, surprisingly, SPIONs treatments markedly corrected the disturbed expression of inflammatory genes and genes controlling mitochondrial biogenesis and functions at mRNA and protein levels in BAT and WAT. The observed effects indicated SPIONs as a powerful inducer of WAT browning and activator of BAT functions where the SPIONs treatment significantly suppressed the markedly enhanced expression and protein level of TNF-α in WAT and BAT. This effect may result from declined leptin secretion, which is supported by the correlation studies which indicate a positive correlation between leptin level and the expression of TNF-α. This effect indicates the anti-inflammatory role of SPIONs in the adipose tissues of obese rats.

Obese rats treated with the doses of the two different coatings of SPIONs alone or in combination with orlistat showed a significant upregulation of PGC-1α, UCP-1, and SIRT-1 expression compared with untreated obese rats in WAT and BAT. Orlistat treatment showed a mild but significant effect on the expressions of these genes. Obese rats treated with combined treatment of orlistat and SPIONs coated with PEG-2000 at the dose of 22 µmol Fe/kg have significant upregulation of PGC-1α expression compared with orlistat-treated rats in both WAT and BAT. This dose showed the highest upregulation effect on PGC-1α expression in WAT, which has a significantly higher expression level compared with control rats. A similar pattern of changes was observed in the mtDNA-CN. Moreover, SPIONs coated with PEG-2000 showed better effects than those coated with PEG-550.

In our present study, the enhanced expression of PGC-1α, which is a central player that regulates the browning program in WAT [57], may cause the enhanced expression of SIRT-1 and UCP-1 in the WAT to be 1.1 and 3.7-fold control values, respectively, and the increased mtDNA-CN in WAT to be higher than the control value. The correlation studies confirm the association between the PGC-1α and WAT browning and BAT activation, as its expression was positively correlated with SIRT-1, UCP-1, and mtDNA-CN and negatively correlated with circulating NEFA. These patterns of gene expression changes may indicate the transformation from WAT into BAT phenotype or browning (or beiging) of the existing WAT. The browning phenomenon has been recognized based on the expression of these specific thermogenic markers that regulate beiging transcription [58]. 

Sirtuin-1’s post-translational modification, such as deacetylation, is a major contributor to the WAT browning [59]. The present data indicated the central role of SIRT-1 in the anti-obesity effects of SPIONs, as it was significantly upregulated in the WAT and BAT of obese rats treated with SPIONs. The correlation studies confirm the critical role of SIRT-1 in the browning of WAT and activation of BAT, as its expression is positively correlated with PGC-1α and UCP-1 expression and negatively correlated with circulating NEFA.

The exact mechanism of the epigenetic effects of SPIONs in vivo is unclear and needs extensive investigations. However, both moieties of SPION-PEG-550 and SPION-PEG-2000 may participate in the observed actions in diabetic rats. PEG moiety facilitates transport across membranes and penetration into intracellular spaces and mitochondria and allows distribution into distant tissues after intraperitoneal injection and exerts significant physiologic effects on the distant organs [60]. 

The exact molecular mechanism(s) involved in the influence of SPIONs on insulin sensitivity is unclear. A few experiments have been conducted to investigate the metabolic effects of SPIONs. Sharifi et al. recorded a decrease in the expression of genes implicated in the growth of obesity and T2D in human primary adipocytes treated with SPIONs [21]. Interestingly, Ali et al. recently reported the potential anti-diabetic effects of SPIONs mediated through correction of hepatic PGC-1α expression and other components of insulin signaling in hepatic tissues and modulation of lipid metabolism and adipocytokines, leptin, and adiponectin [20]. The last study indicated hepatorenal toxicities as a major concern at high doses of SPIONs (44 µmol Fe/kg and 66 µmol Fe/kg) [20]. So, in the present study, we used the low dose (22 µmol Fe/kg) in combination with orlistat to avoid the possible toxicities of the higher dose (44 µmol Fe/kg and 66 µmol Fe/kg), which showed no significant ameliorative effects on AST and ALT activities and even worsened the parameters of the kidney function, urea and creatinine levels, compared with the untreated rats. On the other hand, the low dose alone or in combination with orlistat significantly ameliorates serum activities of AST and ALT compared with the untreated obese rats with no worsened effects on urea and creatinine levels.

### Study Limitations

Nanoparticles biodistribution study is one of the limitations in our study. A systematic study should be performed with the aim to identify their blood circulation half-life time, biodistribution, and clearance. Another limitation is the determination of the principal component in the nanoparticles responsible for this effect and finally determination of the mechanism of action.

## 5. Conclusions

From the results of the present study and the above discussion, for the first time, a promising effect of SPIONs as an anti-obesity agent that is superior to the conventionally used drug orlistat in the HFD rat model has been reported. It was demonstrated that SPIONs influence the expression of genes involved in lipid and glucose metabolism and therefore may be used as therapeutics for the treatment of diabetes and obesity. These effects may be mediated through suppression of WAT expansion, induction of WAT browning, and activation of BAT. The mechanism of action of SPIONs could be mediated through inducing the expression of the thermogenic genes PGC-1α, SIRT-1, and UCP-1 and mitochondria biogenesis in BAT and WAT. SPIONs coated with PEG-2000 are more efficient anti-obesity agents than those coated with PEG-550. The combination of the low dose of SPION-PEG-2000 (22 µmol Fe/kg/week) with daily orlistat has the best efficiency for the treatment of obesity.

## Figures and Tables

**Figure 1 pharmaceutics-14-02134-f001:**
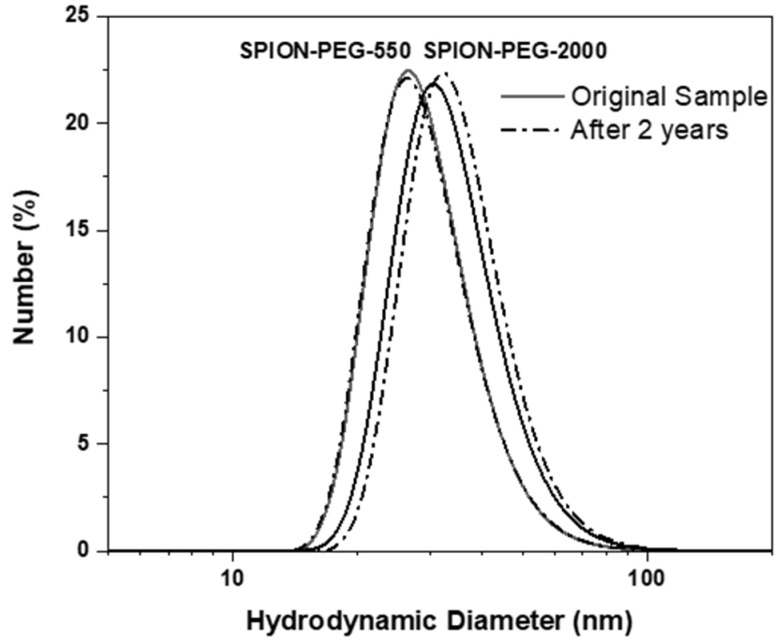
Particle size distribution in water determined by dynamic light scattering (DLS) measurements after being synthesized and after two years of storage.

**Figure 2 pharmaceutics-14-02134-f002:**
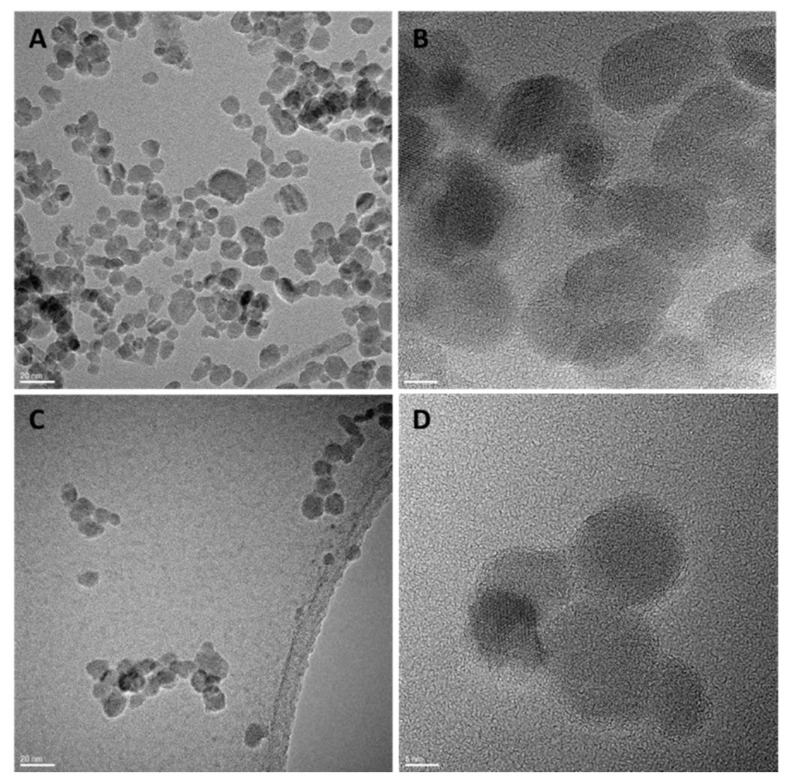
TEM and HR-TEM images of nanoparticles coated with PEG (Mw: 550), (**A**,**B**), and PEG (Mw: 2000), (**C**,**D**), respectively.

**Figure 3 pharmaceutics-14-02134-f003:**
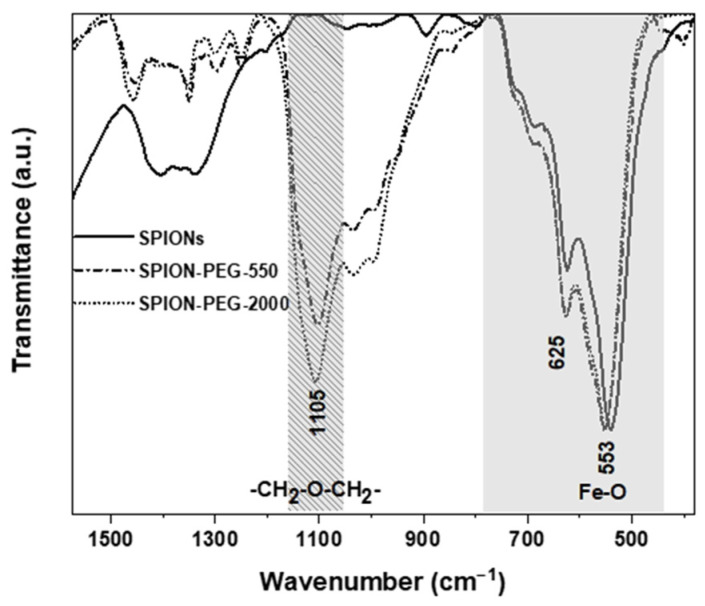
Fourier transform infrared (FTIR) spectra of SPIONs, SPION-PEG-550 and SPION-PEG-2000. The characteristic C-O-C ether stretching vibration band of PEG (1105 cm^−1^) and the bands associated with the Fe-O vibrational modes in γ-Fe_2_O_3_ (625 cm^−1^ and 553 cm^−1^) are highlighted in grey stripped pattern and grey, respectively.

**Figure 4 pharmaceutics-14-02134-f004:**
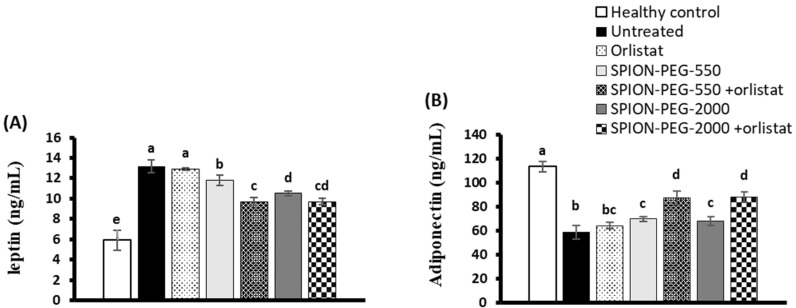
Serum leptin (**A**) and adiponectin (**B**) levels in control rats and obese rats untreated or treated with SPIONs and/or orlistat. Data presented as mean ± SD, and n = 8. Groups were compared at *p* < 0.05 using one-way ANOVA and Tukey post hoc test, and those which are not assigned with a shared letter (a–e) are statistically significant.

**Figure 5 pharmaceutics-14-02134-f005:**
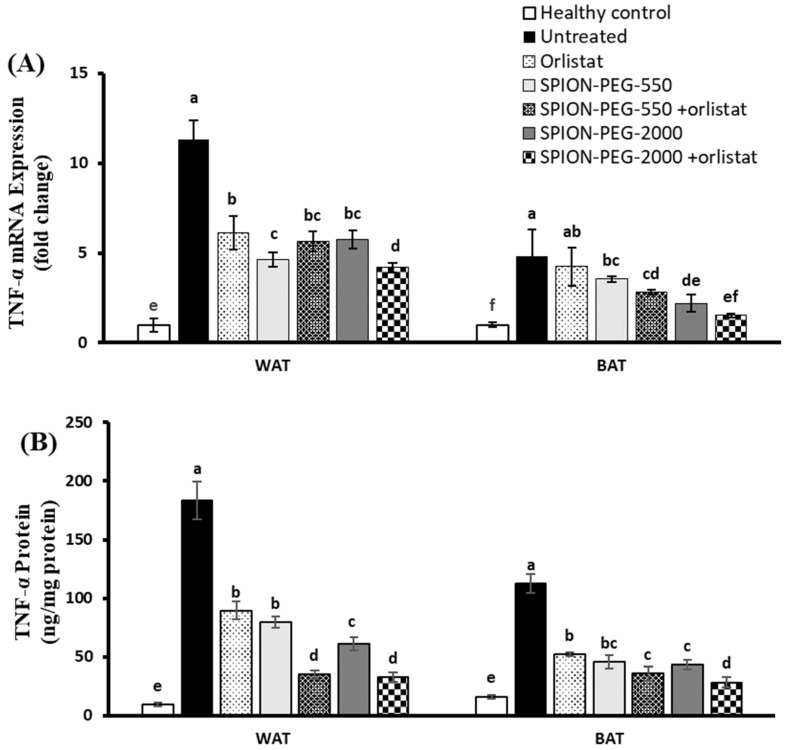
TNF-α expression in white and brown adipose tissues at mRNA (**A**) and protein (**B**) levels in control rats and obese rats untreated or treated with SPIONs and/or orlistat. Data presented as mean ± SD, and n = 8. Groups were compared at *p* < 0.05 using one-way ANOVA and Tukey post hoc test, and those which are not assigned with a shared letter (a–f) are statistically significant.

**Figure 6 pharmaceutics-14-02134-f006:**
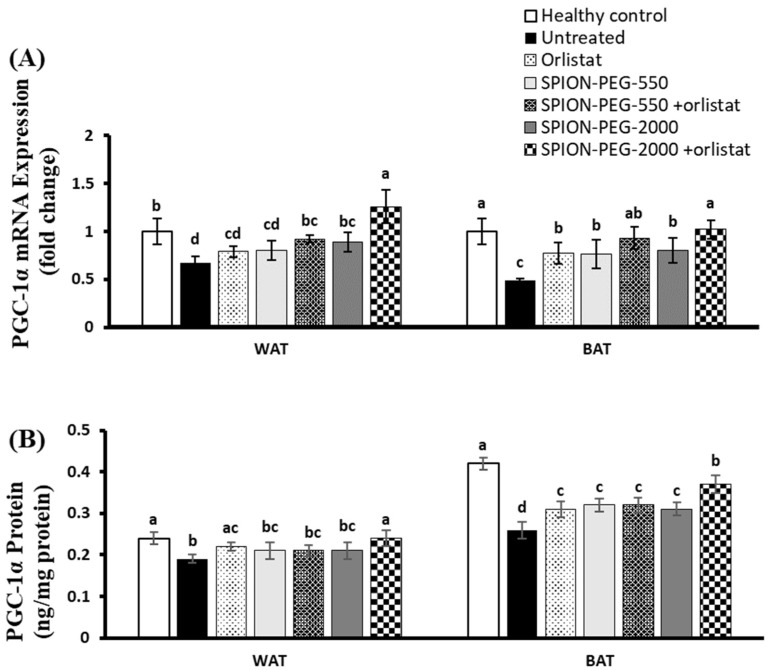
PGC-1α expression in white and brown adipose tissues at mRNA (**A**) and protein (**B**) levels in control rats and obese rats untreated or treated with SPIONs and/or orlistat. Data presented as mean ± SD, and n = 8. Groups were compared at *p* < 0.05 using one-way ANOVA and Tukey post hoc test, and those which are not assigned with a shared letter (a–d) are statistically significant.

**Figure 7 pharmaceutics-14-02134-f007:**
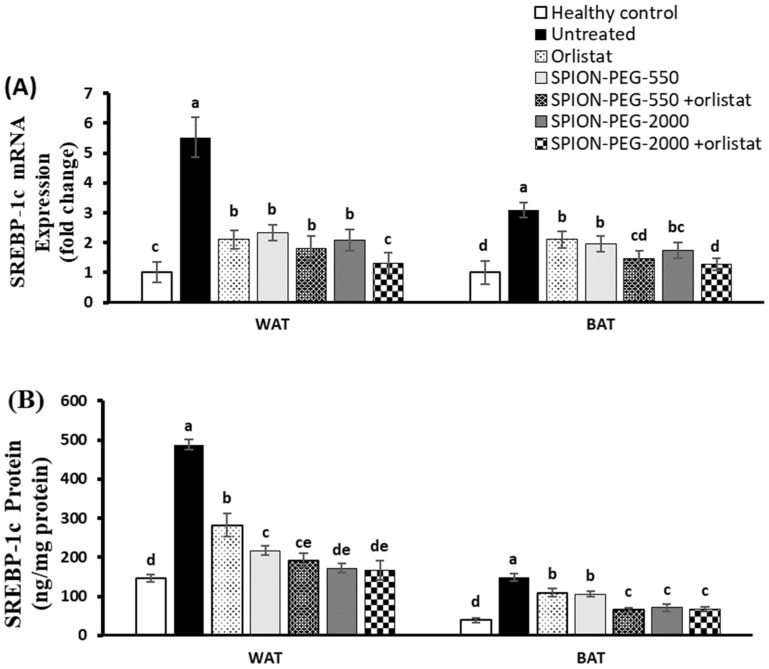
SREBP-1c expression in white and brown adipose tissues at mRNA (**A**) and protein (**B**) levels in control rats and obese rats untreated or treated with SPIONs and/or orlistat. Data presented as mean ± SD, and n = 8. Groups were compared at *p* < 0.05 using one-way ANOVA and Tukey post hoc test, and those which are not assigned with a shared letter (a–e) are statistically significant.

**Figure 8 pharmaceutics-14-02134-f008:**
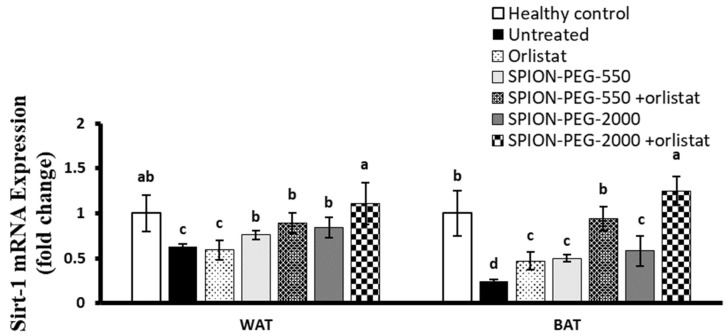
SIRT-1 expression in white and brown adipose tissues at mRNA level in control rats and obese rats untreated or treated with SPIONs and/or orlistat. Data presented as mean ± SD, and n = 8. Groups were compared at *p* < 0.05 using one-way ANOVA and Tukey post hoc test, and those which are not assigned with a shared letter (a–d) are statistically significant.

**Figure 9 pharmaceutics-14-02134-f009:**
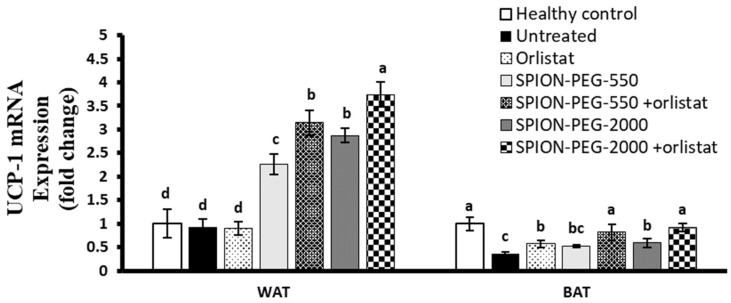
UCP-1 expression in white and brown adipose tissues at mRNA level in control rats and obese rats untreated or treated with SPIONs and/or orlistat. Data presented as mean ± SD, and n = 8. Groups were compared at *p* < 0.05 using one-way ANOVA and Tukey post hoc test, and those which are not assigned with a shared letter (a–d) are statistically significant.

**Figure 10 pharmaceutics-14-02134-f010:**
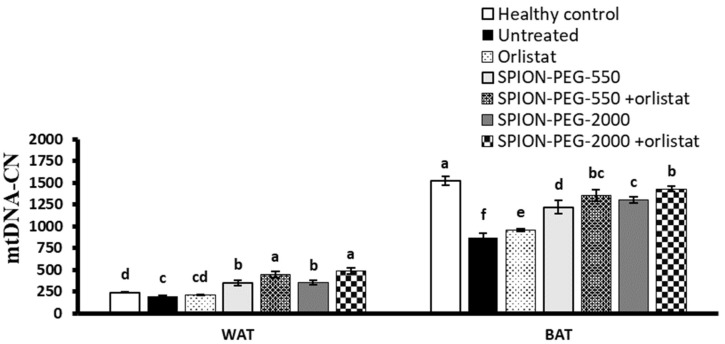
Mitochondrial DNA copy number (mtDNA-CN) in white and brown adipose tissues in control rats and obese rats untreated or treated with SPIONs and/or orlistat. Data presented as mean ± SD, and n = 8. Groups were compared at *p* < 0.05 using one-way ANOVA and Tukey post hoc test, and those which are not assigned with a shared letter (a–f) are statistically significant.

**Table 1 pharmaceutics-14-02134-t001:** Primer sets of the gene expression of PGC-1α, SIRT-1, UCP-1, SREBP-1c, TNF-α, and GAPDH.

Gene	Accession Number	Primer Sequence	
**PGC-1α**	**NM_031347.1**	**F:**	GTGCAGCCAAGACTCTGTATGG
		**R:**	GTCCAGGTCATTCACATCAAGTTC
**SIRT-1**	**NM_001372090.1**	**F:**	TGGCAAAGGAGCAGATTAGTAGG
		**R:**	CTGCCACAAGAACTAGAGGATAAGA
**UCP-1**	**NM_012682.2**	**F:**	AGAGGTGGTCAAGGTCAG
		**R:**	ATTCTGTAAGCATTGTAAGTCC
**SREBP-1c**	**NM_001276708.1**	**F:**	GACGACGGAGCCATGGATT
		**R:**	GGGAAGTCACTGTCTTGGTTGTT
**TNF-α**	**NM_012675.3**	**F:**	GGGCTCCCTCTCATCAGTTC
		**R:**	TCCGCTTGGTGGTTTGCTAC
**GAPDH**	**NM_017008.4**	**F:**	GGGTGTGAACCACGAGAAATA
		**R:**	AGTTGTCATGGATGACCTTGG

**Table 2 pharmaceutics-14-02134-t002:** Statistical analysis of initial and final body weights and weight gain in the different studied groups.

Groups		Initial Weight (g)	Final Weight (g)	Weight Gain (g)
**Healthy** **control**		229 ± 10 ^b^	250 ± 11 ^d^	21 ± 5 ^c^
**Obese rats**	**Untreated**	370 ± 20 ^a^	439 ± 29 ^a^	69 ± 9 ^a^
	**Orlistat**	355 ± 24 ^a^	411 ± 27 ^ab^	56 ± 13 ^ab^
	**SPION-PEG-550**	352 ± 24 ^a^	393 ± 29 ^b^	41 ± 9 ^be^
	**SPION-PEG-550 +orlistat**	357 ± 16 ^a^	363 ± 15 ^c^	6 ± 4 ^cd^
	**SPION-PEG-2000**	357 ± 22 ^a^	389 ± 29 ^b^	32 ± 17 ^ce^
	**SPION-PEG-2000 +orlistat**	354 ± 19 ^a^	367 ± 18 ^c^	13 ± 6 ^c^

Results are expressed as means ± S.D of 8 rats for each group. Groups were compared at *p* < 0.05 using one-way ANOVA and Tukey post hoc test, and those which are not assigned with a shared letter (a–e) in the same column are statistically significant.

**Table 3 pharmaceutics-14-02134-t003:** Statistical analysis of glucose homeostasis parameters in the different studied groups.

Groups		FBG (mg/dL)	Insulin (µIU/mL)	HOMA-IR
**Healthy** **control**		104.5 ± 10.6 ^e^	6.8 ± 0.76 ^c^	1.7 ± 0.14 ^e^
**Obese rats**	**Untreated**	214.3 ± 38.7 ^a^	10.2 ± 1.2 ^a^	5.4 ± 1.4 ^a^
	**Orlistat**	189.3 ± 17.4 ^ab^	9.08 ± 0.58 ^a^	4.2 ± 0.54 ^b^
	**SPION-PEG-550**	180.5 ± 4.2 ^b^	9.5 ± 0.62 ^a^	4.2 ± 0.21 ^b^
	**SPION-PEG-550 +orlistat**	155.6 ± 18.2 ^c^	8.3 ± 0.38 ^b^	3.2 ± 0.28 ^c^
	**SPION-PEG-2000**	169.5 ± 7.3 ^bc^	9.1 ± 0.69 ^a^	3.8 ± 0.33 ^b^
	**SPION-PEG-2000 +orlistat**	123 ± 20.3 ^de^	8.08 ± 0.64 ^b^	2.4 ± 0.32 ^de^

Results are expressed as means ± S.D of 8 rats for each group. Groups were compared at *p* < 0.05 using one-way ANOVA and Tukey post hoc test, and those which are not assigned with a shared letter (a–e) in the same column are statistically significant.

**Table 4 pharmaceutics-14-02134-t004:** Statistical analysis of parameters of liver and kidney function tests in the different studied groups.

Groups		ALT (IU/L)	AST (IU/L)	Urea (mg/dL)	Creatinine (mg/dL)
**Healthy** **control**		36.7 ± 4.3 ^c^	122 ± 12 ^c^	18 ± 3 ^b^	0.66 ± 0.1 ^b^
**Obese rats**	**Untreated**	56 ± 6.2 ^a^	173.1 ±14.1 ^a^	24 ± 3.6 ^a^	0.78 ± 0.05 ^a^
	**Orlistat**	48 ± 3.1 ^b^	154 ± 5.8 ^b^	22 ± 3.2 ^ab^	0.73 ± 0.04 ^a^
	**SPION-PEG-550**	51.2 ± 3.6 ^a^	149.3 ± 5 ^b^	25 ± 3.2 ^a^	0.76 ± 0.07 ^a^
	**SPION-PEG-550 +orlistat**	45.2 ± 4.7 ^bc^	142.7 ± 5.3 ^b^	21 ± 2 ^ab^	0.75 ± 0.04 ^a^
	**SPION-PEG-2000**	48.5 ± 4.5 ^b^	155.1 ± 6.4 ^b^	27 ± 2.6 ^a^	0.72 ± 0.05 ^ab^
	**SPION-PEG-2000 +orlistat**	42.7 ± 3.5 ^bc^	147.3 ± 4.7 ^b^	25 ± 2.4 ^a^	0.77 ± 0.07 ^a^

Results are expressed as means ± S.D of 8 rats for each group. Groups were compared at *p* < 0.05 using one-way ANOVA and Tukey post hoc test, and those which are not assigned with a shared letter (a–c) in the same column are statistically significant.

**Table 5 pharmaceutics-14-02134-t005:** Statistical analysis of lipid profile parameters and NEFA in the different studied groups.

Groups		TG (mg/dL)	TC (mg/dL)	HDL-C (mg/dL)	LDL-C (mg/dL)	NEFA (pg/mL)
**Healthy** **control**		37.6 ± 3.1 ^f^	121 ± 9.2 ^e^	49 ± 2.4 ^a^	64.3 ± 9.6 ^e^	0.44 ± 0.05 ^d^
**Obese rats**	**Untreated**	62.2 ± 3.1 ^a^	168 ± 8.9 ^a^	33 ± 1.3 ^d^	122 ± 8.9 ^a^	1.2 ± 0.06 ^a^
	**Orlistat**	47 ± 2.9 ^c^	145.6 ± 3.1 ^c^	45 ± 2.2 ^ab^	91 ± 4.5 ^c^	0.67 ± 0.03 ^c^
	**SPION-PEG-550**	57.1 ± 2.2 ^ab^	156.2 ± 2.4 ^b^	36 ± 3.5 ^d^	108 ± 4 ^b^	0.85 ± 0.04 ^b^
	**SPION-PEG-550 +orlistat**	46.1 ± 4.1 ^c^	144 ± 4.7 ^c^	44 ± 2.2 ^bc^	91 ± 4.8 ^c^	0.63 ± 0.03 ^c^
	**SPION-PEG-2000**	54±3.4 ^bd^	155 ± 3.9 ^b^	40 ± 3.3 ^c^	103.5 ± 5.3 ^b^	0.81 ± 0.05 ^b^
	**SPION-PEG-2000 +orlistat**	44 ± 3.7 ^ce^	142 ± 4.5 ^cd^	44 ± 2.9 ^bc^	89.3 ± 2.2 ^c^	0.59 ± 0.02 ^c^

Results are expressed as means ± S.D of 8 rats for each group. Groups were compared at *p* < 0.05 using one-way ANOVA and Tukey post hoc test, and those which are not assigned with a shared letter (a–f) in the same column are statistically significant.

**Table 6 pharmaceutics-14-02134-t006:** Correlation studies.

			Leptin Level	NEFA Level	PGC-1α Expression	SIRT-1 Expression	UCP-1 Expression	mtDNA-CN
**Leptin level**	r		_	0.658 *	(WAT) ns	(WAT) ns	(WAT) −0.446 *	(WAT) −0.759 *
					(BAT) −0.401	(BAT) −0.358 *	(BAT) −0.477 *	(BAT) −0.797 *
**PGC-1α** **expression**	WAT	r	ns	−0.577 *	_	0.606 *	0.803 *	0.419 *
	BAT	r	−0.401	−0.499 *	_	0.785 *	0.765 *	0.535 *
**SIRT-1** **expression**	WAT	r	ns	ns	0.606 *	_	0.438 *	ns
	BAT	r	−0.358 *	−0.706 *	0.785 *	_	0.844 *	0.382 *
**UCP-1** **expression**	WAT	r	−0.446 *	−0.69 *	0.803 *	0.438 *	_	0.51 *
	BAT	r	−0.477 *	−0.692 *	0.765 *	0.844 *	_	0.546 *
**SREBP-1c** **expression**	WAT	r	0.41 *	0.547 *	−0.388 *	−0.331 *	−0.599*	−0.403 *
	BAT	r	0.597 *	0.551 *	ns	−0.428 *	−0.418 *	−0.518 *
**TNF-α** **expression**	WAT	r	ns	ns	−0.455 *	−0.533 *	−0.295 *	ns
	BAT	r	0.582 *	0.459 *	−0.343 *	−0.459 *	−0.448 *	−0.562 *
**mtDNA-CN**	WAT	r	−0.759 *	−0.756 *	0.419 *	ns	0.51 *	_
	BAT	r	−0.797 *	−0.613 *	0.535 *	0.382 *	0.546 *	_

Correlation studies obtained by using Pearson correlation test in which **r** = Pearson correlation coefficient and ***** = statistically significant (*p* < 0.005); **ns** means not significant.

## Data Availability

Data will be available by request to the corresponding authors.

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
