# Peer review of "The Anti-Obesity Potential of Superparamagnetic Iron Oxide Nanoparticles against High-Fat Diet-Induced Obesity in Rats: Possible Involvement of Mitochondrial Biogenesis in the Adipose Tissues"

_pharmaceutics, 2022, doi:10.3390/pharmaceutics14102134_

Round 1
Reviewer 1 Report
Dear author’s
I was pleased to review your manuscript «The anti-obesity potential of superparamagnetic iron oxide nanoparticles against high-fat diet-induced obesity in rats: possible involvement of mitochondrial biogenesis in the adipose tissues». The subject is quite interesting. Overall, the paper is well written with good English. The bibliography is complete and useful. The results are clearly presented. However, some aspects need to be pointed.
1. As for me, it is necessary to provide the TEM images of nanoparticles coated with PEG (Mw: 550) and with PEG (Mw: 2000). The authors the authors refer to their paper [19], however, the images shown in the paper have poor resolution or present only 1 particle. The author needs to improve the quality of TEM images with high magnification, besides it is it is important to provide images of a statistically significant number of particles, both types (with PEG Mw: 550 and 2000).
2. Authors claim, that nanoparticles coated by PEG with different molecular weights showed different an anti-obesity activity. What are the reasons for this phenomena? May be this is due to the different amount of PEG on the surface of the particles. I think, it is need use other techniques to characterize the nanoparticles surface (IR-spectroscopy or XPS).
Author Response
Reviewer 1
I was pleased to review your manuscript «The anti-obesity potential of superparamagnetic iron oxide nanoparticles against high-fat diet-induced obesity in rats: possible involvement of mitochondrial biogenesis in the adipose tissues». The subject is quite interesting. Overall, the paper is well written with good English. The bibliography is complete and useful. The results are clearly presented. However, some aspects need to be pointed.
- As for me, it is necessary to provide the TEM images of nanoparticles coated with PEG (Mw: 550) and with PEG (Mw: 2000). The authors the authors refer to their paper [19], however, the images shown in the paper have poor resolution or present only 1 particle. The author needs to improve the quality of TEM images with high magnification, besides it is it is important to provide images of a statistically significant number of particles, both types (with PEG Mw: 550 and 2000).
Response: We would like to thank the reviewer for this comment and following his suggestion, TEM and HR-TEM images for both samples were repeated and provided in Figure 2, section 3.1.
- Authors claim, that nanoparticles coated by PEG with different molecular weights showed different an anti-obesity activity. What are the reasons for this phenomena? May be this is due to the different amount of PEG on the surface of the particles. I think, it is need use other techniques to characterize the nanoparticles surface (IR-spectroscopy or XPS).
Response: We appreciate the comment of the reviewer and agree that a detailed description of the synthetic methodology and characterization of the nanoparticles is required and is important issue especially for biomedical applications. These data were omitted in the present manuscript and was referred to our previous publication, where a complete description of the synthesis and characterization was included both in the main manuscript and in the supporting information. In the particular aspect that the reviewer comments, the presence and quantification of the PEG coating in both samples were carried out by means of FTIR-spectroscopy and thermogravimetric analysis. According to the thermograms obtained, the calculated mass of PEG present in the samples was around 13% for sample SPION-PEG-2000 and 6% for sample SPION-PEG-550. This data is consistent with the presence of polymer chains of higher molecular weight and a higher content in organic polymer in the SPION-PEG-2000 as was confirmed in the FTIR spectra where this sample presents a higher intensity in the bands associated with the polyethylene glycol chains bound to the surface of the iron oxide nanoparticles. Following the reviewer comment these data and Figure 3 showing the FTIR spectra of the SPIONs and the SPIONs-PEG used in this work has been included in Section 3.1.
The anti-obesity activity could be due to the different amount of PEG but we cannot confirm as a systematic study should be carried out to know the principal component responsible for this effect. Therefore, a new section (4.1. Study limitations) was added to the manuscript.
Reviewer 2 Report
The present research article deals with the anti-obesity potential of SIPONs in rats in combination (or not) with the typical anti-obesity drug orlistat. Data provided by authors suggest that combined treatment of both substances (oral dosage of orlistat plus intraperitoneal injection of SPIONs) permits to reduce some of typical obesity-related markers throughout an increment of basal metabolic rate and a transformation of the white adipose tissue into a healthier, more energy-consuming brown tissue.
The manuscript in its current form is very well written and relevant to the current state of the art in obesity. Moreover, claimed conclusions are solidly supported by given experiments. Nevertheless, I would appreciate some additional information/discussion on the following aspects.
Despite SPIONs’ synthesis has been reported in a previous contribution, and thus a detailed synthetic procedure could be omitted, no clue is given on how these particles are prepared for intraperitoneal injection: storage and self-life of colloid, solvent/buffer employed, sterilization process, quantification of iron, etc. Please, give some details on employed methodologies on the experimental section.
As the topic has not been extensively reviewed on the literature, I was not able to find relevant information about the claimed lipotropic effect of SPIONs (line 538); mainly when the outer polar (PEG) layer must provide a hydrophilic behavior. Moreover, I would also appreciate some information on the fate of these SPIONs if available. For instance, any information about how they behave upon injection. Do they manage to dissolve? How was established the weekly dosage?
I understand that the intravenous dosage of SPIONs must somehow induce an effect on adipocytes, browning the white adipose tissue. However, as stated by authors, this process is not fully understood and must be studied in detail. Do you have additional information on this? Did you try the administration of other approved iron oxide formulations (ie. iron gluconate) to determine to which stent the nanoparticulate formulation is responsible of this effect?
It is known that iron cations can induce certain oxidative stress intracellularly due to the catalytic conversion of oxygen into reactive oxygen species. Do you have any information on the possible oxidative stress on the adipose tissue.
Author Response
Reviewer 2
The present research article deals with the anti-obesity potential of SIPONs in rats in combination (or not) with the typical anti-obesity drug orlistat. Data provided by authors suggest that combined treatment of both substances (oral dosage of orlistat plus intraperitoneal injection of SPIONs) permits to reduce some of typical obesity-related markers throughout an increment of basal metabolic rate and a transformation of the white adipose tissue into a healthier, more energy-consuming brown tissue.
The manuscript in its current form is very well written and relevant to the current state of the art in obesity. Moreover, claimed conclusions are solidly supported by given experiments. Nevertheless, I would appreciate some additional information/discussion on the following aspects.
- Despite SPIONs’ synthesis has been reported in a previous contribution, and thus a detailed synthetic procedure could be omitted, no clue is given on how these particles are prepared for intraperitoneal injection: storage and self-life of colloid, solvent/buffer employed, sterilization process, quantification of iron, etc. Please, give some details on employed methodologies on the experimental section.
Response: We would like to thank the reviewer for this comment that helped us to improve our manuscript. Following the reviewer comment an additional paragraphs were added to section 2.1, where detailed descriptions of the methodology are included.
- As the topic has not been extensively reviewed on the literature, I was not able to find relevant information about the claimed lipotropic effect of SPIONs (line 538); mainly when the outer polar (PEG) layer must provide a hydrophilic behavior. Moreover, I would also appreciate some information on the fate of these SPIONs if available. For instance, any information about how they behave upon injection. Do they manage to dissolve? How was established the weekly dosage?
Response: Regarding the the lipotropic action of SPIONs, this effect of SPIONs may be a consequence of the corrected glucose homeostasis and insulin resistance, however, such effect needs further investigation.
Regarding the fate of SPIONs we completely agree on the importance of this study. Therefore, we added a new section (4.1. Study limitations) to the manuscript, in which we mentioned the biodistribution study as one of the limitations in our study. However, similar systems, composed of maghemite nanoparticles embedded in poly(4-vinyl pyridine) (P4VP) and then coated with PEG (Mw 1000 and 200) with hydrodynamic diameter of 163 nm, have shown complete clearance from the circulation 2 h post-intravenous injection and their accumulation in the liver using MRI technique with no toxicity observed in vivo until 60 days post-injection (L.M.A Ali et al. Future Sci OA, 2017;5(1):FSO235). Moreover, using SPIONs (Dh 18 nm) and SPECT technique showed their accumulation in the kidney after 3.5 h post-intravenous injection (V. Gómez-Vallejo et al. Nanoscale, 2018; 10, 14153).
- I understand that the intravenous dosage of SPIONs must somehow induce an effect on adipocytes, browning the white adipose tissue. However, as stated by authors, this process is not fully understood and must be studied in detail. Do you have additional information on this? Did you try the administration of other approved iron oxide formulations (ie. iron gluconate) to determine to which stent the nanoparticulate formulation is responsible of this effect?
Response: In fact, we did not use any approved iron oxide nanoparticles and study their effect in parallel to our system. Hence, we added this observation to section 4.1. Study limitations and we would like to thank the reviewer for this suggestion, which will help us in our investigations in the near future.
- It is known that iron cations can induce certain oxidative stress intracellularly due to the catalytic conversion of oxygen into reactive oxygen species. Do you have any information on the possible oxidative stress on the adipose tissue.
Response: In fact, the role of SPIONs-induced ROS production may play role in their biological effects, however previously (Unpublished) we assessed the markers of oxidative stress in different tissues of animals exposed to SPIONs and find mild but not significant elevation in these markers. So, we can suggest that the SPIONs may induced physiological amount of ROS production that play important role as a cell signalling molecules. Also, the link between SPIONs, mitochondria, and adipocytes must not be ignored